# The Negative Impact of Parental Smoking on Adolescents’ Health-Promoting Behaviors: A Cross-Sectional Study

**DOI:** 10.3390/ijerph18052514

**Published:** 2021-03-03

**Authors:** Mei-Yen Chen

**Affiliations:** 1Department of Nursing, Chang Gung University of Science and Technology, Putz City, Chiayi County 61363, Taiwan; meiyen@mail.cgust.edu.tw; Tel.: +886-(5)-3628800 (ext. 2301); Fax: +886-(5)-3628866; 2School of Nursing, Chang Gung University, Taoyuan 333, Taiwan; 3Department of Cardiology, Chang Gung Memorial Hospital, Chiayi 613, Taiwan

**Keywords:** adolescent, parental smoking, secondhand smoke (SHS), health promotion, rural

## Abstract

The literature has indicated that adolescents’ exposure to secondhand smoke (SHS) and having smoking parents were positively associated with current tobacco smoking. Few studies have explored the association between parental smoking and adolescent’s health-promoting behaviors. This study aimed to examine the prevalence of SHS and the relationship between parental smoking at home and adolescent’s health-promoting behaviors in rural areas. Methods: A school-based and cross-sectional study was conducted from March to December 2019 in western coastal Yunlin County, Taiwan. A total of 1227 adolescents, including 51% boys, anonymously participated in this study. Three parental smoking situations and eight questions of adolescents’ habits extracted from previous studies were applied. A linear regression model was used to analyze the factors associated with adopting health-promoting behaviors. Results: More than half (67.7%) of adolescents experienced parental smoking at home, 90.2% reported their family smoked around them, and 48.8% of participants “sometimes” or “never” avoided SHS. Many participants showed a low frequency of water (49.6%), vegetable (49.1%), and fruit (63.2%) intake, using dental floss (84.7%), and regular exercise (60.6%). The determinants of adolescents’ health-promoting scores were highly associated with avoidance of SHS, less associated with parental smoking, and parents smoking at home and around adolescents. Conclusion: The findings showed that in the rural area, a high prevalence of parental smoking at home and parental smoking around adolescents was noted. It is an important issue that parental smoking was negatively associated with adolescent’s health-promoting behaviors.

## 1. Introduction

The literature indicates that cigarette smoking kills more than seven million people every year around the world [1]. More than six million of these deaths are the result of tobacco use, and nearly one million are nonsmokers who are exposed to secondhand smoke (SHS) [1]. Exposure of adolescents to SHS or tobacco advertisements and having smoking parents were positively associated with current tobacco smoking [2]. The nicotine contained in tobacco is highly addictive and a major risk factor for cardiovascular and respiratory diseases, as well as many different types of cancer caused by vascular disruption, inflammation, and endothelial dysfunction [3,4]. Moreover, SHS exposure has led to adverse health outcomes, causing 1.2 million deaths annually [1]. Nearly half of all children breathe air polluted by tobacco smoke and every year, many children die due to illnesses related to SHS [5]. Furthermore, the use of tobacco products mostly begins from adolescence [6].

In Taiwan, officials reported that the adult smoking rate had declined from 2008 to 2018, from 21.9% to 13%, and the number of adolescent smokers in middle school had reduced from 7.8% to 2.8% [7]. Recently, 25,000 deaths associated with smoking and 3000 caused by SHS were reported. On average, one person dies every 20 minutes in Taiwan due to SHS [7]. Moreover, in disadvantaged areas, the adult smoking rate is still high. According to the regulation law of Taiwan’s Tobacco Hazards Prevention Act [8], smoking is completely prohibited in public places with indoor areas for persons younger than 18 years. It also stated that tobacco products should not be provided for people under 18 years, and parents, guardians, or others in charge of taking care of these adolescents shall forbid them to smoke. However, this law did not include parental smoking at home with children under 18 years. Many adults or parents continued to smoke at home or around youths [9].

The World Health Organization (WHO) states that “adolescents are marked by physiological maturation of all body systems and other life-changing transformations, including from childhood dependence on parents and caregivers to adult independence, this period of life requires tailored health and education services, protection and health promotion designed to be aligned with their developmental stage and to meet their needs” [10]. Several studies indicated that adolescents who regularly exercised consumed an adequate amount of vegetables, fruits, and water, maintained their oral hygiene, including brushing their teeth and using dental floss, and slept for an adequate amount of time and showed a positive correlation with their general health, including physical and mental health [11,12]. For instance, Silk and Kwok [13] pointed out that oral disease can have a profound effect on overall health, including pain, missing school, heart disease, and even death with sepsis through the infection mechanism. Adolescents have specific needs about oral health in addition to the usual lifelong issues of caries management. Dental caries can lead to pain, infection, interference with eating, and worse effects on the eruption of permanent teeth. These manifestations can range from demineralization to loss of tooth structure or destruction of the crown, a process of dynamic and active decay characterized by various periods of destruction and repair [13]. This is preventable by adopting healthy behaviors, such as regular toothbrushing, favorable dietary habits, and regular dental check-ups [12,13].

Many studies on youths’ smoking habits have examined smoking prevalence, ethnic disparities, socioeconomic status, and parental SHS exposure [14,15,16]. However, only a few studies have analyzed the association between SHS from family members’ smoking and adolescent’s health-promoting behaviors [5,9]. Health promotion is defined as a “process of enabling people to increase control over and improve their health” [10]. The first international conference on health promotion was held in Ottawa in 1986 and was primarily a response to growing expectations for a new global public health movement [10]. Most recently, the 9th global conference was titled “Promoting health in the sustainable development goals in the 2030 agenda” [10]. The International Council of Nurses also encouraged nurses to integrate tobacco prevention, cessation, and avoidance of SHS as part of their regular nursing practice [17,18]. In Taiwan, each school should at least employ a nurse. This is a good opportunity for promoting adolescent’s health via health needs assessment and providing further health-related education or counseling. However, few studies have explored the association between parental smoking and adolescent’s health-promoting behaviors in rural areas. Therefore, the purpose of this study was to examine the prevalence of SHS and the relationship between parental smoking at home and adolescent’s health-promoting behaviors in rural areas.

## 2. Materials and Methods

### 2.1. Design, Sample, and Setting

A cross-sectional, school-based, and self-administered anonymous survey was conducted from three rural middle schools between March and December 2019 in coastal, southwestern Yunlin County, Taiwan. Owing to limited resources, the research team has collaborated with the three middle schools since 2014 and held regular annual smoking behavior screenings and provided 150 minutes in three days (50 min/per day) related to the tobacco prevention education after screening for each class in the 7th grade for each school. Three situations of parental smoking at home and a brief form of eight items of adolescent health-promoting behaviors were collected and implemented for 7th to 9th graders aged between 12 and 15 years. In total, census data of 1288 adolescents were included from the three middle schools. Forty-eight students were the school’s athletes who could not join the survey as they were outside the class or campus, and thirteen did not turn in the informed consent forms. After excluding the 61 missing samples, a total of 1227 adolescents participated in this study.

### 2.2. Procedure and Ethical Consideration

The study was approved by the Institutional Review Board (IRB No. 201800428B0). After taking permission from the relevant school administrators, informed written consent was obtained from the adolescents and their guardian(s). The school nurses and physical education (PE) teachers explained the study purpose and procedure to all students before the day of the survey. A pair of two research assistants (senior nursing students) came into each class (the mean class size was around 27–32 students) and gave brief instruction for the 3 situations of parental smoking at home. Students completed the questionnaire anonymously and filled out the scale for about 10–15 minutes.

### 2.3. Measurements

Demographic characteristics. These included sex, age, grade, and smoking status. Regarding smoking status, participants were asked to respond to “Do you have experience in cigarette smoking?” and the answers were categorized as “never, ceased: former user but quit more than six months ago, and current user”.Health-promoting behaviors. These behaviors were determined using eight items of adolescents’ habits extracted from previous studies and works in the literature [11,12]. Participants were asked to respond to the following behaviors with “never/ sometimes” or “usually/always”: (1) Eating breakfast: “Do you eat breakfast every day?”; (2) Water intake: “Do you drink at least 1500cc of water every day?”; (3) Vegetable intake: “Do you consume at least three portions of vegetables every day?” (4) Fruit intake: “Do you consume two portions of fruit every day?” (5) Brushing teeth: “Do you brush your teeth before sleep every day?” (6) Using dental floss: “Do you use dental floss before sleep every day?” (7) Regular exercise: “Do you exercise for 30 minutes, at least three times per week?” and (8) Sleep: “Do you sleep for at least 6 to 8 hours every day?” The responses were categorized as low frequency for never/sometimes, and high frequency for usually/always. Each item recorded 0 and 1 point with low/high frequency. The total health promotion score ranged from 0 to 8.Secondhand smoke (SHS). SHS was determined using three items of parental smoking situations. Participants were asked to respond to the following behavior questions with “yes/ no” or “never/sometimes/usually”, which included: (1) Parental smoking: “Do your parents or family members (e.g., grandparents or relatives) smoke at home every day?” (2) Parental smoking around them: “Do your parents or family members smoke around you every day?” (3) Avoidance of SHS: “How often do you choose to avoid SHS when your parents or family members smoke around you?”

### 2.4. Statistical Analyses

Data analyses were conducted using SPSS 22 (IBM SPSS Inc, Chicago, Illinois). All the tests were two-sided, and *p*-values <0.05 were considered statistically significant. The total health promotion score ranged from 0 to 8. The mean value was 4.46 (standard deviation: 1.66). The skewness and kurtosis were 0.03 and −0.44, respectively, indicating the distribution was generally normal. The difference of total health promotion score in the adolescents with different characteristics was tested using an independent sample t-test (for binary variables) or one-way analysis of variance (for multi-category variables) with Scheffé’s post hoc test. Finally, the associated factors of total health promotion score were determined using a multivariable linear regression analysis which contained the following covariates: gender, parental smoking at home, parental smoking around adolescents, and avoidance of SHS.

## 3. Results

Approximately 51% of the participants were male (*n* = 627), and the mean age was 13.5 years. The distribution of participants in the 7th, 8th, and 9th grade was 34.7%, 31.1%, and 34.2%, respectively. Most of the participants (92.1%) reported that they had never smoked, 6.8% had a previous smoking experience, and 1.1% were current smokers. Table 1 shows that more than half (*n* = 831, 67.7%) were exposed to parental smoking at home, 90.2% (*n* = 750) of parents “sometimes or usually” smoked around the adolescents, and 48.8% (*n* = 405) of participants “never or sometimes” avoided SHS. Regarding health habits, many participants showed a low frequency of inadequate water intake (49.6%), vegetable (49.1%) and fruit (63.2%) intake, using dental floss (84.7%), and regular exercise (60.6%).

In the univariate analysis, Table 2 shows that the total health-promoting score was higher in male adolescents than in female ones. Adolescents with parental smoking at home had significantly lower health-promoting scores than those without. More parental smoking around the adolescents was correlated to a lower total health-promoting score. Adolescents who usually avoided SHS tended to have greater health-promoting scores than those who never or sometimes avoided.

Multivariable regression analysis (Table 3) demonstrates that the determinants of adolescents’ health-promoting scores were avoidance of secondhand smoke (*p* < 0.001), male adolescents (*p* < 0.01), parental smoking (*p* = 0.025), and parents smoking at home and around adolescents (*p* = 0.037).

## 4. Discussion

The findings of the study indicated that only a few participants (*n* = 83, 6.8%) reported having smoking experience (both cigarette and e-cigarette user) and current users were 1.1% (*n* = 14). This result was much lower than some studies conducted in Taiwan and other countries. For instance, Wang et al. [6]. found 24.3% of middle school students had tried a tobacco product, and the smoking rate was 12.5% in the United States and 2.8~7.1% in Taiwan [12,19]. This might be because the research team had initiated a nurse-led school health promotion program focusing on education and prevention of tobacco use for one hour/weekly for three weeks in each class during the first semester for enrolled students since 2014, and the principals of the three schools paid substantial attention to these issues. Therefore, if we want to maintain this low smoking rate, it is necessary to use more innovative strategies to prevent smoking experience and promote SHS education.

However, SHS exposure at home in the present study (67%) was lower than Indonesia (85.4%) [16], but higher than Taiwan (46.7%) [19], Kuwait (45.8%) [20], Macao (41.3%) [9], and also higher than the average of international studies (30.4%) conducted by Veeranki et al. [21] in 168 countries between 1999 and 2008. Exposure to SHS at home was substantially high among adolescents in rural Taiwan. This could be because most of our participants came from lower socioeconomic families, and as the study settings were around rural areas, most families were farmers and fishermen. Previous studies had shown that social disparities in parental smoking and young children’s exposure to SHS at home were correlated [2,5]. Another possible reason might be because parental smoking was measured differently. In this study, while measuring parental smoking, other family members, such as grandparents or relatives who smoked at home, were also included. In rural areas, it is very common for grandparents or relatives to live with their extended family. Living with three generations of their family or relatives, made adolescents be exposed to a higher probability of SHS in the domestic area. Therefore, further studies should consider this phenomenon and provide the skills to avoid SHS and culture-tailored smoking cessation programs.

These findings showed that many participants had a low frequency of inadequate water intake, vegetables and fruit intake, using dental floss, and regular exercise. Moreover, 90% of family smokers “sometimes or usually smoked” around the adolescents, and fewer than half of them “never or sometimes” chose avoidance of SHS. However, if adolescents chose to avoid SHS at home, it was positively associated with better health-promoting behaviors, including eating breakfast, water intake, vegetable and fruit intake, brushing their teeth, using dental floss, and adequate sleep. Furthermore, males had a better score than female adolescents in the aforementioned behaviors and overall health-promoting behaviors, but worse in brushing teeth and using dental floss. This phenomenon was similar to some studies in Spain. Perales-García et al. [22] found less than half of youths had adequate water, and Finnish females between the ages of 12 and 17 years tended to be inactive [23]. As Silk and Kwok [13] mentioned, oral disease can have a profound effect on overall health but is one of the most unmet healthcare needs of adolescents. Teenage years are a higher risk time for oral piercings, increased sugar intake, nicotine initiation, and orthodontic considerations. This is particularly important because lifelong health habits are created during these formative years, and prevention opportunities for caries and periodontitis management are only available at this age [13]. Further studies could consider gender differences when conducting a health-promoting program.

It is crucial to promote health-related behaviors in adolescents and ensure SHS-free homes by school health-promoting programs, especially by school nurses to reach the Sustainable Development Goals [17]. Although Taiwan is the 51st country in the world to accept the WHO Framework Convention on Tobacco Control [12], actions and policies for the prevention of adolescents’ SHS exposure are scarce. It is urgent to advocate an advanced ban for SHS exposure at home for youths and implement innovative smoking education programs, such as empowering adolescents to learn some effective skills to avoid SHS or encourage skills for parental smoking at home.

## 5. Limitations

The present study has some limitations. First, the cross-sectional study design limits the ability to establish causality. Second, the participants were recruited using census data but only in one county, which may limit the ability to generalize the findings. Third, self-completed questionnaires with three situations of parental smoking at home and their health-promoting behaviors could have underestimated its prevalence. Fourth, although a brief form of adolescents’ health-promoting behaviors was anonymously collected, the subjective data may distort the real situation. For instance, the current smoking rate might underestimate the fact. Finally, we did not explore parents’ perceptions regarding the adverse health effects of children’s exposure to SHS. Furthermore, the parent’s socioeconomic status (e.g., parental education, occupation, and income level) might affect both SHS rate and health-promoting behaviors. Thus, further study should consider clarifying the misunderstandings regarding parental smoking and their socioeconomic situation.

## 6. Conclusions

A high prevalence of parental smoking at home, parental smoking around adolescents, and youth not avoiding SHS was found among rural adolescents. Moreover, these factors were negatively associated with adolescents’ health-promoting behaviors. It is an emerging issue for primary healthcare providers and school health nurses to initiate health-promoting programs, including education strategies for youths to avoid SHS and advocate for banning parental smoking at home, especially for a family which has youths younger than 18 years. Furthermore, the inclusion of parents in preventive and health promotion programs, especially in disadvantaged areas, is also an important issue.

## Figures and Tables

**Table 1 ijerph-18-02514-t001:** Demographic characteristics of the participants (*n* = 1227).

Variable	*n* %	Low Never/Sometimes (*n* %)	High Usually/Always (*n* %)
Gender			
Female	600 (48.9)		
Male	627 (51.1)		
Age (mean = 13.5; SD = 1.0; range = 12–17)			
Grade			
7th	426 (34.7)		
8th	381 (31.1)		
9th	420 (34.2)		
Adolescent’s smoking behavior			
Former user (never/ceased)	1213 (98.9)		
Current user	14 (1.1)		
Parental smoking at home No Yes	396 (32.3) 831 (67.7)		
Parental smoking around adolescents (*N* = 831)			
Never Sometime Usually	81 (9.8) 424 (51.0) 326 (39.2)		
Avoidance of secondhand smoke (*N* = 831)			
Never Sometimes Usually	73 (8.8) 332 (40.0) 426 (51.3)		
Eating breakfast		137 (11.2)	1090 (88.8)
Water intake ≥ 1500 cc per day		608 (49.6)	619 (50.4)
Vegetable intake ≥ 3 portions per day		602 (49.1)	625 (50.9)
Fruit intake ≥ 2 portions per day		775 (63.2)	452 (36.8)
Tooth brushing before sleep per day		210 (17.1)	1017 (82.9)
Using dental floss before sleep per day		1039 (84.7)	188 (15.3)
Regular exercise/30 min/per day		744 (60.6)	483 (39.4)
Sleep 6–8 h/per day		232 (18.9)	995 (81.1)

**Table 2 ijerph-18-02514-t002:** Factors associated with adolescents’ health-promoting behaviors (*n* = 1227).

Variable	*n*	Mean	SD	*F*/*t*	*p*-Value	Scheffé’s Post Hoc
Gender				2.77	0.006	
(1) Female	600	4.3	1.6			
(2) Male	627	4.6	1.7			
Parental smoking at home				−4.29	<0.001	
No	396	4.7	1.7			
Yes	831	4.3	1.6			
Parental smoking around adolescents				13.9	<0.001	1 > 2 > 3
(1) Never	477	4.7	1.7			
(2) Sometimes	424	4.4	1.6			
(3) Usually	326	4.1	1.6			
Avoidance of SHS				35.6	<0.001	3 > 1, 2
(1) Never	105	3.7	1.6			
(2) Sometimes	439	4.1	1.6			
(3) Usually	683	4.8	1.7			

SD, standard deviation; SHS, secondhand smoke.

**Table 3 ijerph-18-02514-t003:** Determinant factors associated with adolescent’s health-promoting behaviors (*n* = 1227).

Variables	Unstandardized B	SE	Beta	*t* Value	*p*	95% CI *
Avoidance of secondhand smoke (1 = often)	0.72	0.09	0.22	7.68	<0.001	0.54 to 0.90
Gender (1 = male)	0.32	0.09	0.10	3.47	0.001	0.14 to 0.50
Parental smoking at home (1 = yes)	−0.24	0.11	−0.07	−2.24	0.025	−0.45 to −0.03
Parents smoking around adolescents (1 = often)	−0.24	0.12	−0.06	−2.09	0.037	−0.46 to −0.02

* Confidence interval.

## Data Availability

The datasets analyzed during the current study are not yet publicly available but are available from the corresponding author on reasonable request.

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
