# Peer review of "The Negative Impact of Parental Smoking on Adolescents’ Health-Promoting Behaviors: A Cross-Sectional Study"

_ijerph, 2021, doi:10.3390/ijerph18052514_

Round 1
Reviewer 1 Report
Review Report ijerph-1083383
This is a manuscript that explores the contribution of parental smoking on adolescents’ health promoting behaviours. It is based on a school-based cross-sectional study of the responses from 1227 adolescents aged 12-15 years attending 3 middle-schools in Yunlin County in Taiwan.
It is a nicely written paper but I have some methodological concerns that the author needs to address before I can recommend this for publication.
Major comments:
- The author states that 61 participants were excluded from the final analysis – could they provide a description of the demographic characteristics of this subset and how these may differ, or not, from the included sample. There is potential for exclusion bias.
- In Measurements the author describes the questions for the health promoting behaviours but I do not understand if the responses to these questions are made on scales or are dichotomous responses. If scales were used, how were the data aggregated into High/Low. These details need to be clarified.
- How was the “Health promoting score” used in the regression models calculated? What was the distribution of the scores? Were the results normally distributed?
- Statistical analyses. The author has reported a large number of univariate analyses, stratified by the demographic characteristics and exposure to SHS. I think these results are misleading as these models are not adjusted for any confounders. In addition, stratified models should be accompanied by interaction analyses to assess if the covariate is actually acting as an effect modifier/moderator of the outcome. These interaction analyses should be adjusted for any confounders and key independent determinants of the outcome. Also there are p-values reported for these univariate analyses but I cannot determine what statistical test was used to derive these – are these indicating differences between the categories within the covariate? This requires revision and clear rationale for their choices of statistical approach.
- Limitations: The author has not discussed limitations associated with response bias associated with self-completed questionnaires; selection bias associated with recruitment of participants; and the representativeness of the sample. In the discussion they compare the prevalence of SHS exposure with other settings but I don’t have confidence in the comparability between studies. Can they provide additional information to back up these comparisons.
Minor comments:
- Some minor English grammar correction is needed.
- The Introduction is quite disjointed and requires further work to establish the importance of the research question.
- WHO World Health Organization should be spelled out in full when using the acronym the first time.
- Given the audience for IJERPH – the author needs to explain what ‘caries’ is for a non-oral health audience.
- Death associated with poor oral health is rare, particularly among adolescents. However, I understand that it is a chronic disease that impacts on morbidity which eventually contribute to premature mortality. This could be written more clearly.
- Consistent use of the acronym SHS would improve the manuscript.
Author Response
Response to Reviewer 1 can be found in the attachment.

Reviewer 2 Report
The paper is well-written with the analysis well-described (though there are a few areas in which the English could be improved slightly).
The multivariate analysis described in Table 3 is likely to be confounded by socioeconomic status (SES), which will likely affect both rates of SHS exposure and of other health-promoting behaviours. The suggested direct link between parental smoking at home and general health-promoting behaviour is spurious, and the causal claim in the paper’s title is unfounded.
Line 178: Substantially different rates of smoking from government statistics (ref 12) – I can’t identify the stats in ref 12, when are they from? Is it plausible that the presence of nurses in schools distributing and collecting surveys could have affected children’s responses, which they may not have felt were truly anonymous?
In general areas of the results seem unnecessary. Table 2 doesn’t seem connected to the main point of the paper – is it of note that seventh graders are less likely to brush their teeth frequently than ninth graders, and is it connected to the main point of the paper? Furthermore, given the large number of univariate analyses conducted on the data set, coincidentally “significant” p-values below 0.05 could be expected without underlying significance.
Author Response
Response to Reviewer 2 can be found in the attachment.

Round 2
Reviewer 2 Report
Thanks to the authors for addressing my concerns.
Author Response
Regarding reviewer 2 for English language and style are fine/minor spell check required.
Thanks for your comments, please refer to the following minor spell checks in red.
- P1, Line 13: “rural areas” replace “the rural area”
- P1, Line 20: “low-frequency of” replaces “low frequency”
- P1, Line 20: remove “inadequate”
- P2, Line 87: “health-promoting” replaces “health promoting”
- P2, Line 90: “health-promoting” replaces “health promoting”
- P3, Line 131: “recorded” replaces “recoded”
- P4, Line 162: “health-promoting” replaces “health promoting”
- P4, Line 164: “health-promoting” replaces “health promoting”
- P4, Line 164: “parental” replaces “the parental”
- P4, Line 165: “health-promoting” replaces “health promoting”
- P4, Line 166: “health-promoting” replaces “health promoting”
- P6, Line 210: “vegetables” replace “vegetable”
- P6, Line 216: “health-promoting” replaces “health promoting”
- P6, Line 246: “health-promoting” replaces “health promoting”
Also, the full text font has been consistently changed to “Palatino Linotype”.